# Organoid Technology and Its Role for Theratyping Applications in Cystic Fibrosis

**DOI:** 10.3390/children10010004

**Published:** 2022-12-20

**Authors:** Jessica Conti, Claudio Sorio, Paola Melotti

**Affiliations:** 1Department of Medicine, Division of General Pathology, University of Verona, 37134 Verona, Italy; 2Cystic Fibrosis Centre, Azienda Ospedaliera Universitaria Integrata Verona, 37126 Verona, Italy

**Keywords:** CFTR, cystic fibrosis, organoids, theratyping, personalized medicine

## Abstract

Cystic fibrosis (CF) is a autosomal recessive, multisystemic disease caused by different mutations in the CFTR gene encoding CF transmembrane conductance regulator. Although symptom management is important to avoid complications, the approval of CFTR modulator drugs in the clinic has demonstrated significant improvements by targeting the primary molecular defect of CF and thereby preventing problems related to CFTR deficiency or dysfunction. CFTR modulator therapies have positively changed the patients’ quality of life, especially for those who start their use at the onset of the disease. Due to early diagnosis with the implementation of newborn screening programs and considerable progress in the treatment options, nowadays pediatric mortality was dramatically reduced. In any case, the main obstacle to treat CF is to predict the drug response of patients due to genetic complexity and heterogeneity. Advances in 3D culture systems have led to the extrapolation of disease modeling and individual drug response in vitro by producing mini organs called “organoids” easily obtained from nasal and rectal mucosa biopsies. In this review, we focus primarily on patient-derived intestinal organoids used as in vitro model for CF disease. Organoids combine high-validity of outcomes with a high throughput, thus enabling CF disease classification, drug development and treatment optimization in a personalized manner.

## 1. Introduction

Until a few years ago, treatments for Cystic Fibrosis (CF) were mainly based on relieving symptoms: physiotherapy to enhance airway clearance and combat lung infections and inflammation, nutritional status management and, in case of end-stage lung disease, lung transplantation. The fundamental current standards for CF therapy include pancreatic enzyme supplementation, fat-soluble vitamins and high-calorie ingestion to minimize pancreatic insufficiency and intestinal malabsorption, anti-inflammatory drugs, antibiotics and mucolytics [1,2,3]. Over the last decade, new therapeutic strategies have been proposed and a new class of drugs named CFTR modulators has been included in the therapeutic care of patients that are currently qualified for treatment. The development of new CF therapies has brought benefits in preventing disease complications, improving individual patient well-being and increasing survival rates. In fact, until the 1960s, CF was a fatal and incurable disease in infancy and today most people with CF are reaching adulthood. Pediatric mortality was dramatically reduced and the survival of CF patients has continuously improved with many individuals living up to 40–50 years in some countries today. Indeed, despite care being based on well-established guidelines, there are many health status disparities among CF patients according to healthcare systems, adherence to therapies, treatment type (route of administration, duration, number of daily medications, etc.), as well as patient socio-personal characteristics and genetic background [4]. Poor treatment adherence has been reported in CF and may lead to worse health outcomes and greater healthcare use. Because each patient is different in terms of lifestyle and social and economic aspects, individual motivational support and personalized educational/training courses can help the patient understand the importance of adhering to the therapeutic regimen to obtain the best clinical benefits. Developing a stronger relationship between patients, families/caregivers, and clinical CF researchers could be the first step to improving the therapeutic compliance of patients and relatives, especially critical in the case of children, and setting up a patient-oriented research infrastructure that promotes the translation of research into clinical impact.

The extensive knowledge obtained in this field has greatly modified the practices of care and outlook for CF pediatric patients. During the last 10 years, the use of newborn bloodspot screening (NBS) for the early diagnosis of children with CF has become widely adopted. Earlier diagnosis and CFTR-targeted therapies have led to efficient improvement of the quality of life. The increasing availability of CFTR-targeted drugs that may halt or severely reduce the disease progression and potentially interrupt the pathological sequences leading to CF organ complications provides the rationale for proposing early treatment (including during pregnancy) to reduce or prevent long-term consequences of the disease. As a result, a larger number of CF-affected children could start a treatment path even a few weeks after the birth, a well before showing symptoms or irreversible organ damage. Indeed, children who had access to earlier diagnosis, which means earlier access to medical management and intervention, show better health outcomes at an older age when compared to those children who had a later CF diagnosis [5,6]. The evidence supporting the clinical benefits of NBS programs have been extensively reviewed [7,8,9,10,11].

Furthermore, Szczesniak et al. [12] showed how lung function decline can be used as a parameter to identify individuals who have a higher potential to obtain an advantage whether from new therapies or those already available. For the possibility that the clinical manifestation of CF could be prevented, modulator therapy is increasingly used in younger children and even infants [13,14,15,16]. Over the last decade, significant efforts into high-throughput screening (HTS) of small molecule libraries have enabled the identification of CFTR modulators. CFTR modulator drugs have been described for the first time in 2003 [17]. Ivacaftor (Vertex Pharmaceuticals, MA, US) received a marketing authorization valid throughout the EU on 23 July 2012 (FDA approval on 31 January 2012) thus opening a new era in the treatment of this severe disease. The CFTR modulators currently available in clinic for CF are: ivacaftor, lumacaftor/ivacaftor, tezacaftor/ivacaftor, elexacaftor/tezacaftor/ivacaftor (Vertex Pharmaceuticals, MA, USA) and are currently revolutionizing the management of patients with CF, particularly those with at least one F508del variant (up to 85% of patients worldwide). These drugs primarily target CFTR variants that present a gating defect (class III variants) or a processing defect (Class II variants), but data in vitro and in vivo indicate how these drugs can be effective in other types of variants that affect CFTR function and/or processing. Table 1, Table 2, Table 3 and Table 4 summarize the current status of indications from the Food and Drugs Administration (FDA) approved CFTR modulators and the array of CFTR variants for which they are approved.

Since 2018, when the combinations Tezacaftor/Ivacaftor and Tezacaftor/Ivacaftor/Elexavaftor were marketed and introduced into clinical practice, the number of mutations that are responsive to modulators has steadily increased. Figure 1 shows how many mutations are targeted by a single CFTR modulator or by their combination. All variants initially approved for Ivacaftor treatment subsequently became eligible for treatment with Ivacaftor combined with one or two correctors. To date, approximately 50% of the mutations identified as responsive to modulators can be targeted by Ivacaftor or the combinations Tezacaftor/Ivacaftor and Tezacaftor/Ivacaftor/Elexacaftor. The number of responsive variants approved for Symkevi and Kaftrio are 2 and 32 respectively. On the basis of the clinical picture and the tolerability of the patient, it is the clinician’s responsibility to choose the therapy.

Alongside conventional therapies, CFTR modulators represent an important advance in the management of CF, as instead of treating the consequences of CFTR dysfunction, they target the underlying cause associated with CFTR mutations [18].

Nevertheless, the broad range of CFTR mutation classes with various intracellular consequences, epigenetics, modifying genes, individual responsiveness/tolerance to drugs, and still unknown reasons lead to a huge variability in the clinical phenotype of CF. Indeed, CF shows a huge variability between patients, for which several possible explanations can be proposed. Firstly, in the CFTR gene itself, over 2100 CFTR variants have been reported in the Cystic Fibrosis Mutation Database (accessed on 31 October 2022: https://cftr2.org/). Only ~20% of them are proven as CF disease-causing variants. For those mutations for which the functional defect has not yet been thoroughly studied prediction of the clinical manifestations is particularly challenging [19,20]. Secondly, other variables include genetic modifiers that regulate CFTR or other genes’ function (in epithelial or non-epithelial cells), and interactions with the environment [21,22]. As a consequence, phenotypic variations and variability in response to CFTR modulators can be observed even in patients with identical CF mutations. For this reason, each CF patient is unique in terms of response, compliance and tolerance to drugs and disease progression. Although it is clear that in vitro systems cannot recapitulate the complex interactions occurring between drugs and the whole organisms, nevertheless the availability of a patient’s avatar reproducing at least its genomic background at the cellular level represent a significant step forward in this direction. The available data suggest indeed that the capability of the channel to respond to agonist/s and the efficacy of modulators in correcting the molecular defect can be properly investigated using any of these models.

The main treatment barrier for CF is to predict the drug response of patients because of genetic complexity and heterogeneity. The most frequent mutations in large groups of patients have been well described with regards to clinical effects and CFTR-targeted treatment options associated with them but this is not the case for rare, orphan mutations identified in only a few patients worldwide, described in such a small patient population that classical clinical trial studies are not feasible [19]. Therefore, there is an urgent need to evaluate the individual drug response in patient-derived model systems that reflect pharmacological treatment efficacy in vivo. The ultimate goal is to avoid a try-and-error approach for expensive treatments with potential side effects.

## 2. Alternatives to Conventional Clinical Trials

Since developing a specific drug for each of the CF causing variants is infeasible, other than probably not even necessary, they have been classified according to their molecular mechanisms of the defects and their response to modulators [23,24]. Through a personalized medicine approach, it has been possible to prescribe already commercially available drugs to patients with less common CF mutations, considering that patients with mutations belonging to the same group can be treated by the same therapeutic scheme. Due to the theratyping process and considerable progress in the treatment options, a larger population of individuals with CF (Table 2, Table 3 and Table 4) may benefit from the drug and potentially be cured, although there is still a long way to go for ultra-rare and orphan mutations. Updated information can be retrieved at https://www.vertextreatmentshcp.com/eligibility-tool.

Considering the current high price of CFTR-targeting molecules and variability in the clinical phenotype of CF, even in patients with identical or similar genotypes, it would be ethically more correct to test the efficacy of drugs before administering them to the patient [25] because there are people who benefit from the drug, but there are also individuals who do not obtain an advantage and even those who have adverse effects. To identify effective treatments and avoid undesirable effects, an emerging alternative to genotype-based drugs in CF is personalized/precision medicine, i.e., to determine whether rare CFTR mutations respond to existing (or new) CFTR modulators by pre-evaluating them directly on the patient’s tissues ex vivo, which is now also termed theratyping. This approach proposes the advantage of directly selecting the best drug, or drug combination, for that individual and his/her combined mutations by pre-assessing the efficacy of CFTR-target drugs directly on specimens obtained from CF subject and as a consequence could diminish the approval of treatments that could be ineffective or harmful and create a high expectation to the patient and family members. Predictions based on patient-derived materials as a starting point are nonetheless a more achievable approach than the clinical trial of each CFTR modulator in an N-of-1 trial [26].

Children have the same right to evidence-based therapy as adults but data extrapolation from adults may be inappropriate or misleading [27], therefore it is important to elaborate well-accepted patient-oriented research tools to predict CF treatment response in the pediatric population as well.

## 3. Cell Models for Studying CF Disease Pathogenesis and Therapy

Experimental models based on the use of in vitro cell cultures have allowed for obtaining many and key information on the biological activity of the CFTR protein and its molecular defects, moreover they have permitted the screening of molecules with different pharmacological activities and to evaluate their pharmacological effects suggesting that cell response in vitro could be predict the clinical impact.

Immortalized epithelial cell lines such as A549, BEAS-2B, Calu-3, CFBE41o- and 16HBE14o-, are suitable models, easy to culture and expand [28,29,30]. However, these immortalized cell lines are derived from lung tumor cells or have been transformed and therefore lack original lung cell characteristics and have some disadvantages due to immortalization strategy that can induce genetic instability, karyotype anomalies and altered gene expression [31,32]. Of note is the fact that drug therapy recovery intervention of CFTR mutations is greatly affected by the cell background [33,34,35].

Fischer Rat Thyroid (FRT) cells that ectopically express CFTR cDNA are the pre-clinical, high-throughput model that has been mostly used to successfully develop CFTR modulators. The recent FDA approval for label extension of ivacaftor and ivacaftor/tezacaftor/elexacaftor to patients with different mutations was based primarily on laboratory evidence of efficacy in FRT cells [36,37]. However, this cellular model has intrinsic limitations: FRT cells were developed from Fischer rat thyroid gland, as such its protein folding machinery is not human and this condition might affect the response to treatment [34]. It is equally clear that the same model cannot be minimally predictive of variations in intronic sequences, and that the transfection under an exogenous promoter might alter a proper protein level and processing. As a consequence, the observed effects might not closely match the in vivo situation.

To evaluate individual CFTR modulators’ responses, several assays using CF patient-derived materials have been implemented and are widely used [38,39,40,41,42,43]. Ex-vivo individual-derived specimens, such as human bronchial epithelia (HBE), human nasal epithelial (HNE), intestinal organoids (OGs) and nasal as well as lung spheroids resemble parental organ epithelium morphology and functionality and reflect the complete genetic background of the subjects. These features permit us to come closer to evaluating the response in individual genetic backgrounds and are expected to better predict the clinical effectiveness of the given treatment.

Primary HBE cells are typically obtained by invasive procedures (bronchoscopy or lung transplantation) from lungs with advanced/end-stage disease that may or may not reflect cells’ behavior in early disease. They are usually available in a limited number of severely ill patients so cannot be used for large-scale or theratyping studies.

HNE cells seem to be a good surrogate for human bronchial epithelial cells. They are collected by minimal-invasive procedures such as nasal brushing or scraping of the lower turbinates. The current gold standard for modeling the primarily affected CF lung epithelium is air–liquid interface (ALI) culture of human nasal epithelial cells [39]. However, this nasal cell culture approach has some limitations: it requires a high number of cells, lengthy differentiation protocols, cells have limited ability to expand and HNE-derived cells are not necessarily representing the features of lower airways [44].

Nevertheless, in the last few years, several research groups have explored several approaches that allow for isolation, expansion, and differentiation of primary nasospheroids [40,45,46,47,48]. Recently a standardized protocol was proposed [49]. Nasal brush biopsies collection from infants through to adults is well known across most centers and can be performed with risk comparable to that of a nasopharyngeal swab for virus detection. In addition, the demonstration of the correlation between CFTR modulator responses in nasal and intestinal OGs provides early evidence that CFTR functional assay in nasal airway OGs can also be used to predict modulator efficacy in a genotype-dependent manner [48,50,51].

Another tissue representative of CF disease is the gastrointestinal tract which is affected in utero or early after birth by diseases such as meconium ileus and pancreatic insufficiency, the latter featuring typical pancreatic cysts after which the disease was named. Interestingly when nasal and intestinal mucosa were compared as in vivo biomarkers of CFTR function to distinguish people with CF (pwCF) and healthy controls, Intestinal Current Measurement (ICM) was found superior to Nasal Potential Difference (NPD) and ICM demonstrated substantially greater power than NPD to detect low levels of residual CFTR function [52,53] suggesting a potential superiority of intestinal over nasal mucosa for theratyping applications. In 2009, Sato et al. developed the basis for intestinal organoid technology [54] that recapitalized in-vivo tissue architecture forming three dimensional structures that can develop in a crypt-like epithelium [55]. In 2011, human intestinal OG cultures were described by the same group [56]. Organoid culture protocol requires a delicate balance of several growth factors such as Wnt, R-spondin and Noggin plus a specific basement membrane matrix. Intestinal OGs could be greatly expanded in vitro over long periods without losing their stemness and biobanked for future use without a need for genetic modifications or further patient inconvenience for repeated biopsies [56]. Human intestinal OG can be grown from intestinal crypt fragments isolated following a rectal biopsy procedure that causes only limited discomfort to patients being painless, usually well accepted by patients and feasible in people of all age groups (including newborns) without a need for anesthesia/sedation [57]. ICM in rectal biopsies have been included for decades in the diagnostic algorithm for CF and CFTR related disorders [58,59], in particular to aid establish or refute a diagnosis of CF in patients with equivocal sweat test or genetic testing results [60], and in many cases rectal samples can be used for generation of rectal OGs after ICM.

Intestinal OGs remain the most advanced three-dimensional in vitro model for CF to date. Other than being a primary target organ in CF it is worth noting that CFTR represents the dominant channel responsible for ion and fluid secretion in gastrointestinal cells, which make intestinal OGs valuable models to investigate CFTR function and modulation [61,62,63]. Moreover, while the airways are significantly affected, the intestine is not significantly affected by chronic inflammation and infection with CF pathogens. Furthermore, the lack of significant chronic organ damage and remodeling in the intestine is a factor that reduces the chance to have CFTR channel function affected independently of the presence of CFTR variants [64]. Finally, intestinal OGs develop fast from the biopsy which results in a shorter time for readout, likely derived from the exceptionally high cell turnover in the intestinal epithelium that renews itself within 3–5 days.

## 4. Disease Modeling of Intestinal OG in Infants and Children

In vitro human intestinal OGs provide a unique development model that can be applied to study intestinal prematurity diseases such as necrotizing enterocolitis, short bowel syndrome, Hirschsprung’s disease, infectious disease in the intestine and genetic diseases such as CF [65].

Intestinal OGs are robust tools for studying genetic diseases due to their genetic and phenotypic stability such as CF and cancer where genetics can influence disease severity, prognosis, and drug efficacy [66,67].

Certainly, research in the CF field provides an appropriate example where OG technology has generated a relevant impact. Here, patient-derived intestinal OGs have been used for disease modeling, drug screening, and personalized medicine and have provided advantages in the development of CFTR-modulating drugs. Intestinal OGs have demonstrated high validity and high-throughput potential to predict drug efficacy in individuals with CF [68,69].

The use of OG in high-throughput screening enables pre-clinical testing of many compounds on patient-specific tissues. In vitro tests based on patient-derived rectal OGs can facilitate rapid access to new treatment or can assist in the identification of variants that are currently not registered for treatment but can potentially be responsive to CFTR-targeting drugs. This general approach is underway within the HIT-CF program (e.g., Human Individualized Therapy: HIT-CF program in Europe, www.hitcf.org) using patient-derived rectal OG to assess cellular responses to various CFTR modulators. This European project aims to develop personalized treatments for PwCF allowing those with rare CFTR mutations access to treatment.

The potential of intestinal OGs as a pre-clinical model and for personalized medicine has been demonstrated in different studies [70,71,72]. The first study to directly compare OG response and clinical phenotype prospectively included 34 newborns with CF. Newborns were clustered into low responders or high responders. Low response in OG was related to increased pulmonary and pancreatic disease parameters at the age of 1 year. In CF children, intestinal OG response corresponds with clinical phenotypes at 1 year of age as well as in vivo sweat Cl^-^ concentration (SCC) and ICM. Interestingly, in cases where SCC and ICM disagreed, FIS appeared to correctly align with the clinical indicators allowing the in vivo residual CFTR function to be accurately estimated for each patient [73].

Notably, Berkers et al. reported a strong correlation between in vivo and in vitro response of CFTR modulators indicating how OG can play a key role in personalized medicine approach [74]. The predictive value of intestinal OG response was demonstrated to significantly correlate with the most important therapeutic endpoints: ICM, reduction in SCC, and improvements in lung function measured as the volume of air that can forcibly be blown out in one second, after full inspiration and expressed as predicted percent FEV1 (ppFEV1) after administering the CFTR modulator therapies to patients. In addition, in vitro CFTR modulator responses in OG displayed excellent accuracy for stratifying drug responders from non-drug responders [68,74] suggesting the organoid model is suitable for guiding label extension or compassionate use for PwCF with rare genotypes [71,75].

Data from Pranke et al., also indicate that in vitro analysis of nasal epithelium may correlate with in vivo outcomes [76] and airway epithelium cultures obtained from nasal brushing of a CF child have been demonstrated to be a useful model for theratyping [77,78]. Although bronchoalveolar lavage has been proposed for this purpose, a simple nasal brushing is a more easily performed, well tolerated, and minimally invasive technique. It is therefore possible to study upper airway epithelial cells in very young CF infants to obtain information on the state of inflammation, infection, CFTR conductance and modulator drugs’ responsiveness [79]. Whether they represent a faithful model for lower airway epithelia, and whether this truly represents a requisite for theratyping applications require further studies [44].

## 5. CFTR Bioassay in Intestinal Organoids

Predicting individual patient response necessary to optimize the application of a given treatment, based on the use of currently available and future CFTR modulators, require a robust and standardized pre-clinical test [80].

The majority of data describing the readout for quantifying the CFTR function and how it can be recovered by CFTR modulators in intestinal organoids derived from the Forskolin-Induced Swelling test (FIS). This test was described first by Dekkers and collaborators in 2013 [43]. Exposing intestinal OG to Forskolin causes the cells to rapidly increase their levels of cyclic adenosine monophosphate (cAMP), resulting in the opening of the CFTR channel. As chloride ions move through the channel and into the lumen, the OGs increase in size due to luminal water intake. The FIS phenotype is absent in human and mouse OG lacking functional CFTR gene products (e.g., two disease-causing mutations or CFTR knockout) and is inhibited by chemical CFTR inhibitors, supporting full CFTR-dependency of the FIS readout [43]. The FIS assay allows for considerable throughput since initially it had been set up in 96-well plates [43], but recently it was developed into a 384-well-based high-throughput screening format.

FIS quantitatively correlates with CFTR function and genotype: healthy control OGs appear already swollen and a further increase in swelling appear marginal due to the presence of liquid in the lumen that cannot accumulate over physiological limits. Absent FIS is present in patients with CFTR null alleles, is decreased in variants classified as mild, and is strongly reduced in OG derived from severe variants [81,82].

FIS of patient-derived OGs is very helpful to measure patient-specific CFTR activity, comparing CFTR function between individuals presenting with different and even identical CFTR mutations, and quantifying individual CFTR modulator response. Some studies found, however, that in healthy control OG FIS rate is negatively affected by fluid-filled lumens before forskolin treatment leading to an underestimation of CFTR function. The OG initial starting area must be similar to obtain accurate and reliable swelling results. Therefore, to compare healthy donor-derived OG and CF OGs, another test was introduced: the steady-state lumen area (SLA) assay in which lumen surface area is measured as a percentage of total OG size [68]. Whereas CF OGs have SLA between 0% and 10% of the total OG area, healthy control SLA is between 40% and 80%. Steady-state differences in luminal OG area exist between healthy control and CF OGs independent of forskolin. Healthy control rectal OGs have large fluid-filled lumens, suggesting the presence of functional CFTR and physiological cAMP-dependent signaling during standard culture conditions resulting in the luminal transport of salt and fluid. CF OHs have limited luminal volume or do not have lumens that are easily recognizable during visual inspection.

FIS and SLA are complementary assays. FIS of CF and healthy controls are not directly comparable therefore it is more suited to only compare CF conditions. SLA facilitates comparison between CF and healthy control OG but has a limited resolution to discriminate at lower CFTR function levels associated with severe CF disease.

Despite these useful features, a limitation of FIS assay is that the selective delivery of compounds to the apical or basolateral compartment in OG growing as 3D structures is complicated. Apical stimulation can be performed in OG by microinjection, but this is especially challenging for CF OGs because of their limited luminal volume. More recent research has described protocols to generate two-dimensional monolayers grown on porous membrane filters from dissociated 3D OG [83]. A monolayer of intestinal OGs provides easy access to the apical side and allows for the assessment of CFTR rescue and function by traditional electrophysiological measurements in Ussing chambers [84,85]. Furthermore, Ussing chamber measurements in 2D monolayers allow us to separately measure CFTR-mediated Cl^−^ and HCO_3_^-^ transport [85]. This is of great importance as modulator drugs capable of restoring CFTR folding have demonstrated a different impact on CFTR-dependent Cl^-^ transport compared to HCO_3_^-^, suggesting the occurrence of different behavior of rescued CFTR on different genotypes that might differentially affect chloride and bicarbonate transport [86,87]. This observation might have an impact on the predicted in vivo efficacy of a given modulator in the specific CFTR variant and cannot be detected by FIS assay which is unable to distinguish whether the swelling is associated with chloride only or other anions.

Epithelial monolayers of intestinal OG may be a valuable tool to evaluate ion transport of different channels/transporters in different culture conditions that can be precisely modified for the purpose other than assisting in diagnosis and precision medicine testing: Indeed, CFTR-dependent intestinal epithelial ion transport measured on rectal organoid-derived monolayers of subjects carrying distinct CF genotypes correlates well with donor-matched native ICM and FIS of 3D intestinal OG [88]. The high dynamic range of the response of monolayers derived from intestinal organoids demonstrated suitable to identify none/very low to high residual CFTR function and WT-CFTR currents might be used as a reference value for comparing the efficacy of modulator drugs on various CFTR variants [85].

Another CFTR bioassay with intestinal OG that can be used as a complementary diagnostic test is rectal organoid morphology analysis (ROMA). ROMA is based on the measurement of two morphological parameters of the organoids: (1) the intensity ratio (IR) that evaluates the presence or absence of a central lumen, and (2) the circularity index (CI) that measures the roundness of the organoids [89]. As outcomes, the test allows for discrimination of CF patients from healthy subjects and it evaluates if there is a response of OG to modulator treatments. The functional recovery of the CFTR protein is reflected in a more open central lumen and a more circular shape, measured by IR and CI respectively. Current strategies for theratyping are summarized in Figure 2.

## 6. Limitations of Organoid-Based Assays

The most feasible personalized biomarker platforms for testing CFTR function and CFTR modulator responsiveness in the foreseeable future will likely be intestinal and nasal cells and derived OG as they represent key targets and easily accessible sites for cell procurement, with the former currently representing a more robust and reproducible platform based on available data in the literature [90]. In addition to the general limitations of any single tissue-based organoid assay (e.g., lacking host environment-related signal factors) already discussed in previous reviews [91,92], additional specific limitations are currently associated with technical issues such as the need to develop common Standard Operating Procedures (SOPs) among laboratories, the need to train highly skilled personnel and the costs associated to tissue procurement, processing and analysis other than the need to introduce common platforms permitting reference samples and data exchange among laboratories. Rather important is the development of quality control, standardization, and compliance with technical standards possibly via certification. The recognition of the use of primary cells as suitable biomarkers for theratyping applications might eventually be required by certifying agencies, such as the European Medicines Agency (EMA) and the FDA, to enter the approval procedure of future medications or to include new CFTR variants in the approved list. All of these issues are summarized in Table 5.

## 7. Conclusions

The common goal is to give new hope to all PwCF by changing the general perception of CF as a lethal disease. Prospects aim to identify new CFTR modulators and to extend eligibility for drugs now adopted for clinical use to patients with CFTR variants without approved treatment so that all CF patients can eventually be treated with a matched therapy. All these new perspectives will hopefully bring existing and novel CFTR modulators faster to patients with rare mutations and in the long run also bring the most efficacious drug to individual patients who carry more common mutations. The possibility to develop and biobank patient’s “avatars”, such as OGs, that can be developed at birth, recovered, and analyzed once new drugs might become available (a highly likely possibility in the foreseeable future) without the need for resampling ensures previously unthinkable opportunities to match the treatment with the individual molecular asset and develop a much needed theratyping approach, especially for those patients whose variants are so rare to require an N-of-1 approach.

It remains to establish the relationship between the results obtained from patient-derived materials and the long-term results, but as with any new approach, these data will only be collected as they become available.

## Figures and Tables

**Figure 1 children-10-00004-f001:**
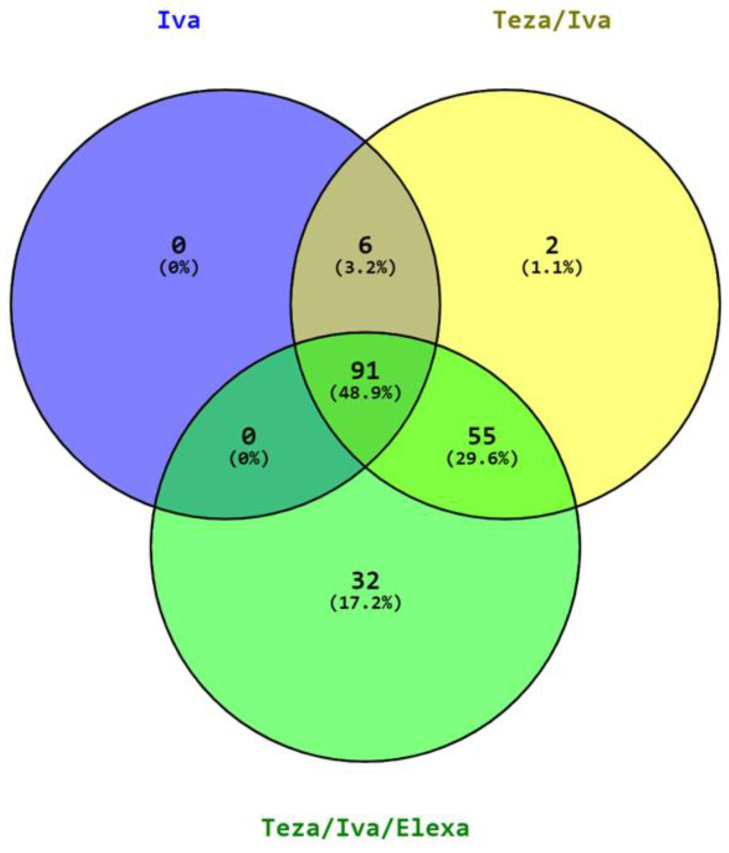
Venn Diagram showing the number and relative percentage of mutations identified so far as responsive to Ivacaftor (Iva) or to the combinations of the correctors Tezacaftor (Teza) and Elexacaftor (Elexa) with the potentiator (Iva).

**Figure 2 children-10-00004-f002:**
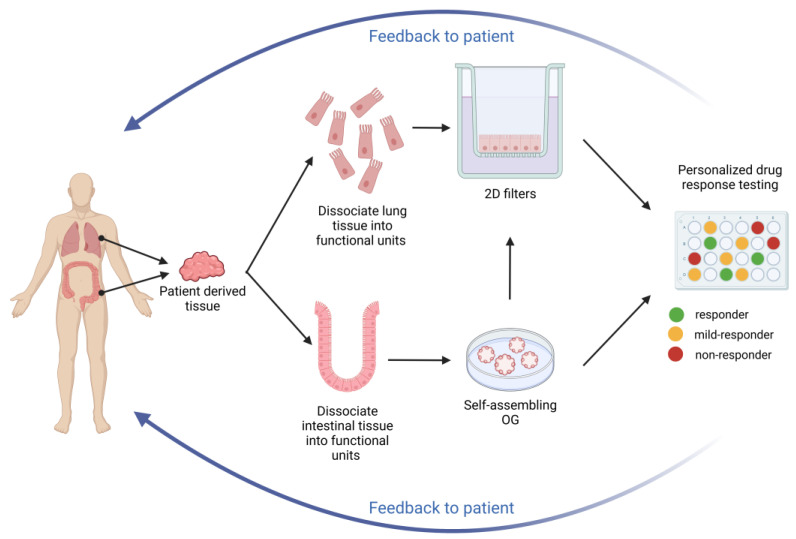
Theratyping strategies. Nasal and intestinal biopsies can be collected and utilized to derive long-term cultures that can be utilized for functional assays measuring CFTR function, as described in the text. The results can inform the clinician on the response of the specific CFTR variants and combinations expressed by the proband thus providing directions on the choice of the most appropriate treatment.

**Table 1 children-10-00004-t001:** Summary of licensed CFTR modulator approved by FDA (https://www.vertexgps.com, accessed on 31 October 2022); worldwide data are available from https://news.vrtx.com, accessed on 31 October 2022.

Modulator	License Age and Characteristics	Mutations
**Ivacaftor**	≥4 months;≥5 kg	Class III gating mutations (Table 2)
**Lumacaftor/Ivacaftor**	≥1 year	Homozygous F508del
**Tezacaftor/Ivacaftor**	≥6 years	Homozygous F508del or at least one copy of responsive mutations (Table 3)
**Ivacaftor/Tezacaftor/Elexacaftor**	≥6 years	At least one F508del mutation or at least one copy of responsive mutations (Table 4)

**Table 2 children-10-00004-t002:** List of CFTR mutations eligible for the treatment with Ivacaftor and approved by the FDA. Available from https://www.vertexgps.com, accessed on 14 November 2022.

711 + 3A→G	D1152H	G194R	I807M	Q237H	R553Q	S1159F
2789 + 5 G→A	D1270N	G314E	I1027T	Q359R	R668C	S1159P
3272–26A→G	E56K	G551D	I1139V	Q1291R	R792G	S1251N
3849 + 10kbC→T	E193K	G551S	K1060T	R74W	R933G	S1255P
A120T	E822K	G576A	L206W	R75Q	R1070Q	T338I
A234D	E831X	G970D	L320V	R117C	R1070W	T1053I
A349V	F311del	G1069R	L967S	R117G	R1162L	V232D
A455E	F311L	G1244E	L997F	R117H	R1283M	V562I
A1067T	F508C	G1249R	L1480P	R117L	S549N	V754M
D110E	F508C/S1251N	G1349D	M152V	R117P	S549R	V1293G
D110H	F1052V	H939R	M952I	R170H	S589N	W1282R
D192G	F1074L	H1375P	M952T	R347H	S737F	Y1014C
D579G	G178E	I148T	P67L	R347L	S945L	Y1032C
D924N	G178R	I175V	Q237E	R352Q	S977F	

**Table 3 children-10-00004-t003:** List of CFTR mutations eligible for the treatment with Tezacaftor/Ivacaftor and approved by the FDA. Patients should carry F508del mutation in both alleles or at least one copy of the mutations listed here. Available from https://www.vertexgps.com, accessed on 14 November 2022.

546insCTA	D1152H	G126D	I601F	P5L	R334L	S912L
711 + 3A→G	D1270N	G178E	I618T	P67L	R334Q	S945L
2789 + 5 G→A	E56K	G178R	I807M	P205S	R347H	S977F
3272–26A→G	E60K	G194R	I980K	Q98R	R347L	S1159F
3849 + 10kbC→T	E92K	G194V	I1027T	Q237E	R347P	S1159P
A120T	E116K	G314E	I1139V	Q237H	R352Q	S1251N
A234D	E193K	G551D	I1269N	Q359R	R352W	S1255P
A349V	E403D	G551S	I1366N	Q1291R	R553Q	T338I
A455E	E588V	G576A	K1060T	R31L	R668C	T1036N
A554E	E822K	G576A/R668C	L15P	R74Q	R751L	T1053I
A1006E	E831X	G622D	L206W	R74W	R792G	V201M
A1067T	F191V	G970D	L320V	R74W/D1270N	R933G	V232D
D110E	F311del	G1069R	L346P	R74W/V201M	R1066H	V562I
D110H	F311L	G1244E	L967S	R74W/V201M/D1270N	R1070Q	V754M
D192G	F508C	G1249R	L997F	R75Q	R1070W	V1153E
D443Y	F508C/S1251N	G1349D	L1324P	R117C	R1162L	V1240G
D443Y/G576A/R668C	F508del	H939R	L1335P	R117G	R1283M	V1293G
D579G	F575Y	H1054D	L1480P	R117H	R1283S	W1282R
D614G	F1016S	H1375P	M152V	R117L	S549N	Y109N
D836Y	F1052V	I148T	M265R	R117P	S549R	Y161S
D924N	F1074L	I175V	M952I	R170H	S589N	Y1014C
D979V	F1099L	I336K	M952T	R258G	S737F	Y1032C

**Table 4 children-10-00004-t004:** List of CFTR mutations eligible for the treatment with Ivacaftor/Tezacaftor/Elexacaftor and approved by the FDA. Patients should carry at least one copy of the mutations listed here. Available from https://www.vertexgps.com, accessed on 14 November 2022.

3141del9	E193K	G551D	I980K	P574H	R352W	S1255P
546insCTA	E403D	G551S	I1027T	Q98R	R553Q	T338I
A46D	E474K	G576A	I1139V	Q237E	R668C	T1036N
A120T	E588V	G576A/R668C	I1269N	Q237H	R751L	T1053I
A234D	E822K	G622D	I1366N	Q359R	R792G	V201M
A349V	F191V	G628R	K1060T	Q1291R	R933G	V232D
A455E	F311del	G970D	L15P	R31L	R1066H	V456A
A554E	F311L	G1061R	L165S	R74Q	R1070Q	V456F
A1006E	F508C	G1069R	L206W	R74W	R1070W	V562I
A1067T	F508C/S1251N	G1244E	L320V	R74W/D1270N	R1162L	V754M
D110E	F508del	G1249R	L346P	R74W/V201M	R1283M	V1153E
D110H	F575Y	G1349D	L453S	R74W/V201M/D1270N	R1283S	V1240G
D192G	F1016S	H139R	L967S	R75Q	S13F	V1293G
D443Y	F1052V	H199Y	L997F	R117C	S341P	W361R
D443Y/G576A/R668C	F1074L	H939R	L1077P	R117G	S364P	W1098C
D579G	F1099L	H1054D	L1324P	R117H	S492F	W1282R
D614G	G27R	H1085P	L1335P	R117L	S549N	Y109N
D836Y	G85E	H1085R	L1480P	R117P	S549R	Y161D
D924N	G126D	H1375P	M152V	R170H	S589N	Y161S
D979V	G178E	I148T	M265R	R258G	S737F	Y563N
D1152H	G178R	I175V	M952I	R334L	S912L	Y1014C
D1270N	G194R	I336K	M952T	R334Q	S945L	Y1032C
E56K	G194V	I502T	M1101K	R347H	S977F	
E60K	G314E	I601F	P5L	R347L	S1159F	
E92K	G463V	I618T	P67L	R347P	S1159P	
E116K	G480C	I807M	P205S	R352Q	S1251N	

**Table 5 children-10-00004-t005:** Limitations and possible solutions to challenges faced with organoid cultures and theratyping approach.

Criticisms	Solutions	Refs
Limited FIS use for high-throughput drug screening	384-well plates format	[40]
Limited FIS of OG with fluid-filled lumens	SLA assay	[68]
Difficult to inject drugs into lumen of 3D organoids Measure CFTR mediated Cl− and HCO3- transport separately	2D monolayers of intestinal OG	[83,84,85]
Technical issues of OG and theratyping (common SOPs and exchange platforms, highly skilled personnel, costs, etc.)	In progress	
Certification of quality control and compliance to technical standards	In progress	
Certifying agencies not accepted CFTR testing in OG	In progress	

## Data Availability

All datasets utilized are indicated throughout the text.

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
