# Peer review of "Organoid Technology and Its Role for Theratyping Applications in Cystic Fibrosis"

_children, 2022, doi:10.3390/children10010004_

Round 1

Reviewer 1 Report

Cystic Fibrosis represents a therapeutic challenge due in part to the disease complexity itself added to the patient heterogenity. Recently, several drugs have been suggested as CFTR modulators on specific responsive patients. Therefore, this review contributes importantly with interesting discussions about state-of-the-art approaches in the field of Personalized Medicine. Remarkably, ex vivo models (theratyping) such patient-derived intestinal or respiratory organoids are on the scope. Authors described in detail several applications, advantages and disadvantages of each ex vivo model. However, the current version of the manuscript could be improved by addressing some minor concerns below:

1. Tables 2-4 could be graphically represented as a Venn Diagram, in order to highlight those mutations targeted by different FDA-approved drugs.

2. In section 3, authors could prepare a Figure displaying the described differences among isolation, expansion, and differentiation of respiratory and intestinal spheroids as explained in the cited protocols by different groups. This will add to the discussion on the most robust and/or reproducible approaches using organoid derivation.

3. In order to summarize the explanations and criticism from authors on  sections 5 and 6, a table displaying the main physiological and clinical advantages and disadvantages of intestinal and respiratory organoids should be structured. This will highlight the open questions and limitations of theratyping models against CF.

I hope my suggestions help improving the quality and significance of this manuscript.

Author Response

Cystic Fibrosis represents a therapeutic challenge due in part to the disease complexity itself added to the patient heterogenity. Recently, several drugs have been suggested as CFTR modulators on specific responsive patients. Therefore, this review contributes importantly with interesting discussions about state-of-the-art approaches in the field of Personalized Medicine. Remarkably, ex vivo models (theratyping) such patient-derived intestinal or respiratory organoids are on the scope. Authors described in detail several applications, advantages and disadvantages of each ex vivo model. However, the current version of the manuscript could be improved by addressing some minor concerns below:

  1. Tables 2-4 could be graphically represented as a Venn Diagram, in order to highlight those mutations targeted by different FDA-approved drugs.

Done

  1. In section 3, authors could prepare a Figure displaying the described differences among isolation, expansion, and differentiation of respiratory and intestinal spheroids as explained in the cited protocols by different groups. This will add to the discussion on the most robust and/or reproducible approaches using organoid derivation.

Actually the protocols are quite similar for a general audience and technical details seems to us to be out of the scope. We rather propose to the reviewer figure 2 that summarizes of the approach and, in our opinion,  might better suit the general audience. Advice and suggestions are highly appreciated.

  1. In order to summarize the explanations and criticism from authors on  sections 5 and 6, a table displaying the main physiological and clinical advantages and disadvantages of intestinal and respiratory organoids should be structured. This will highlight the open questions and limitations of theratyping models against CF.

Done, see table 5

I hope my suggestions help improving the quality and significance of this manuscript.

They surely did, thank you for the suggestions

Reviewer 2 Report

The review is thorough and well organized, and will be a useful educational source for researchers and clinicians. Only minor changes are requested.

1. "Cystic Fibrosis" does not need to be capitalized in the title.

2. The formatting in line 21 is separating the "o" of "outcomes" from the rest of the word.

3. In Table 1, it is mentioned that Ivacaftor is prescribed for patients with Class III gating mutations, but such mutations are not adequately defined elsewhere. It is unlikely that researchers or clinicians unfamiliar with CF would know about gating mutations.

4. In line 194, the abbreviation for organoids (OG) is used, but the abbreviation is not declared until line 207.

5. In line 196, the abbreviation for gastrointestinal (GI) is not declared. Please double check that all acronyms are declared at the first usage.

6. In line 219, ICM does not need to be declared a second time, as it was already declared in line 200.

7. In line 273, ppFEV1 is mentioned as a metric for lung function. Please define and explain this measurement more thoroughly.

8. There are instances where verbs do not match the singular or plural state of the following pronouns or nouns. For example, in line 286: "...whether this study truly represent a requisite...". It should read "whether this study truly represents...". There are many instances of this throughout the manuscript.

Author Response

The review is thorough and well organized, and will be a useful educational source for researchers and clinicians. Only minor changes are requested.

  1. "Cystic Fibrosis" does not need to be capitalized in the title.

fixed

  1. The formatting in line 21 is separating the "o" of "outcomes" from the rest of the word.

fixed

  1. In Table 1, it is mentioned that Ivacaftor is prescribed for patients with Class III gating mutations, but such mutations are not adequately defined elsewhere. It is unlikely that researchers or clinicians unfamiliar with CF would know about gating mutations.

We added a brief description of the principles in lanes 101-104 avoiding to add too many details that can be easily retrieved by interested readers at the suggested links.

  1. In line 194, the abbreviation for organoids (OG) is used, but the abbreviation is not declared until line 207.

fixed

  1. In line 196, the abbreviation for gastrointestinal (GI) is not declared. Please double check that all acronyms are declared at the first usage.

fixed

  1. In line 219, ICM does not need to be declared a second time, as it was already declared in line 200.

fixed

Done (first time now in line 506)

  1. In line 273, ppFEV1 is mentioned as a metric for lung function. Please define and explain this measurement more thoroughly.

Done, (see lane 608-610)

  1. There are instances where verbs do not match the singular or plural state of the following pronouns or nouns. For example, in line 286: "...whether this study truly represent a requisite...". It should read "whether this study truly represents...". There are many instances of this throughout the manuscript.

We checked throughout the manuscript and corrected these unmatched sentences